# Paracetamol (Acetaminophen) and the Developing Brain

**DOI:** 10.3390/ijms222011156

**Published:** 2021-10-15

**Authors:** Christoph Bührer, Stefanie Endesfelder, Till Scheuer, Thomas Schmitz

**Affiliations:** Department of Neonatology, Charité—Universitätsmedizin Berlin, 13344 Berlin, Germany; stefanie.endesfelder@charite.de (S.E.); till.scheuer@charite.de (T.S.); thomas.schmitz@charite.de (T.S.)

**Keywords:** paracetamol, acetaminophen, attention deficit hyperactivity disorder, autism spectrum disorder

## Abstract

Paracetamol is commonly used to treat fever and pain in pregnant women, but there are growing concerns that this may cause attention deficit hyperactivity disorder and autism spectrum disorder in the offspring. A growing number of epidemiological studies suggests that relative risks for these disorders increase by an average of about 25% following intrauterine paracetamol exposure. The data analyzed point to a dose–effect relationship but cannot fully account for unmeasured confounders, notably indication and genetic transmission. Only few experimental investigations have addressed this issue. Altered behavior has been demonstrated in offspring of paracetamol-gavaged pregnant rats, and paracetamol given at or prior to day 10 of life to newborn mice resulted in altered locomotor activity in response to a novel home environment in adulthood and blunted the analgesic effect of paracetamol given to adult animals. The molecular mechanisms that might mediate these effects are unknown. Paracetamol has diverse pharmacologic actions. It reduces prostaglandin formation via competitive inhibition of the peroxidase moiety of prostaglandin H2 synthase, while its metabolite *N*-arachidonoyl-phenolamine activates transient vanilloid-subtype 1 receptors and interferes with cannabinoid receptor signaling. The metabolite *N*-acetyl-p-benzo-quinone-imine, which is pivotal for liver damage after overdosing, exerts oxidative stress and depletes glutathione in the brain already at dosages below the hepatic toxicity threshold. Given the widespread use of paracetamol during pregnancy and the lack of safe alternatives, its impact on the developing brain deserves further investigation.

## 1. Introduction

Paracetamol (acetaminophen, *N*-acetyl-*para*-aminophenol) is among the most popular painkillers used by mothers during pregnancy [1] and by young children worldwide. It is also used to treat fever and has been advocated for pharmacological closure of a patent ductus arteriosus in preterm infants [2,3]. Until recently, paracetamol had been considered safe for use in pregnancy. However, there is mounting (albeit controversial) evidence that it may have long-term negative effects on the offspring when used by pregnant women, increasing the risks for attention deficit hyperactivity disorder (ADHD) and autism spectrum disorder (ASD). This narrative review presents retrieved data and views which at times are difficult to reconcile but open avenues to further research.

## 2. Pharmacology of Paracetamol

### 2.1. Inhibition of Prostaglandin Synthesis

Despite its popularity and use for many years, the safety of its application and its mechanism of action are not fully understood. Paracetamol is a manifold drug, and several complex metabolic pathways are involved in its antipyretic and analgesic action (see schematic overview in Figure 1 and Figure 2). Some of the effects of paracetamol are mediated by reduced prostaglandin formation [4] via competitive inhibition of the peroxidase moiety of prostaglandin-endoperoxide synthase, also called prostaglandin H2 (PG H2) synthase [5,6]. Prostaglandin-endoperoxide synthase is a bifunctional enzyme that consists of a cyclooxygenase (COX) site and a peroxidase site that work in series. The COX site oxidizes arachidonic acid to prostaglandin G2 (PG G2). PG G2 is then rapidly converted by the peroxidase site to PG H2, which goes on to serve as a substrate for several isomerases/synthases that ultimately result in the release of biologically active compounds such as thromboxane A2, prostaglandin I2, or prostaglandin E2 [7]. The COX site can be inhibited by non-steroidal anti-inflammatory drugs such as ibuprofen or indomethacin, while paracetamol acts as a reducing co-substrate of the peroxidase site, lowering the rate of conversion of PG G2 to PG H2 [8].

There are two prostaglandin-endoperoxide synthase isoenzymes, formerly called COX1 (constitutively expressed) and COX2 (inducible). Paracetamol is a partially selective COX2 inhibitor; concentrations of paracetamol necessary to achieve 50% inhibition of the prostaglandin-endoperoxide synthase activity by the inducible isoenzyme (26 µM) are approximately 25% of that by the constitutively expressed isoenzyme (114 µM) [9]. Paracetamol has little anti-inflammatory effect [10] because it inhibits intracellular prostaglandin-endoperoxide synthase but not molecules released from damaged cells [11]. Of note, COX2 knockout mice display autism-related behavior [12].

### 2.2. Interaction with Central Receptors Involved in Nociception

An additional mechanism has been proposed to mediate paracetamol-induced central analgesia and lowering of body temperature. While paracetamol itself acts as an antagonist of transient vanilloid-subtype 4 receptor (TRPV4) [13], the paracetamol metabolite *N*-arachidonoyl-phenolamine (AM404) activates transient vanilloid-subtype 1 receptors (TRPV1) [14] and transient receptor potential ankyrin 1 (TRPA1) [15]. AM404 is generated by de-acetylation of paracetamol to *p*-amino-phenol (in the liver) and subsequent conjugation with arachidonic acid by the enzyme fatty acid amide hydrolase (FAAH; in brain and spinal cord) [16,17]. AM404 can be detected in cerebrospinal fluid after administration of paracetamol [18] and mediates central analgesia by increasing local concentrations of γ-amino-butyric acid (GABA), glutamate, and endocannabinoids, thereby decreasing the connectivity of cortex, amygdala, hypothalamus, and periaqueductal grey [19]. Synaptic endocannabinoid availability is achieved by AM404-mediated inhibition of the anandamide membrane transporter, while AM404 itself acts as a weak agonist of the cannabinoid receptors type 1 and 2. The paracetamol metabolites *N*-acetyl-*p*-benzo-quinone-imine (NAPQI) and *p*-benzoquinone (*p*-BQI) generated by the CYP450 isoform CYP2E1 expressed in brain and spinal cord [20] is also a direct stimulator of TRPA1 [21], but this interaction is limited by its short half-life.

### 2.3. Pharmacokinetics and Toxicology

Paracetamol is mainly excreted following conjugation with glucuronic acid or sulfate. A variable fraction, however, is oxidized in the liver by a number of CYP450 isoforms (CYP2E1, CYP1A2, CYP3A4, and CYP2A6) to NAPQI. NAPQI is a highly reactive compound neutralized by reduced glutathione (GSH), resulting in the generation of L-cysteinyl-S-acetaminophen. If concentrations of NAPQI exceed the available GSH, NAPQI wreaks havoc by covalently binding to thiol groups of various cellular proteins and lipids. The involvement of mitochondria triggers an oxidative stress cascade that leads to accumulation of reactive oxygen species, formation of peroxynitrite, mitochondrial membrane permeability transition, and ultimately cell death [22]. Acute paracetamol toxicity may be antagonized by restoration of GSH stores via early administration of high-dose *N*-acetyl-cysteine administration [23]. At a later stage, moderate hypothermia to induce RNA-binding motif protein 3 [24] appears to be a promising strategy that is awaiting clinical evaluation.

Metabolization of paracetamol to NAPQI occurs mostly in hepatocytes, and acute liver failure constitutes the principal cause of death following intentional (suicidal) or unintentional (unsupervised prolonged administration) paracetamol overdose. Encephalopathy seen in this situation is therefore attributed to liver failure. However, NAPQI is also generated in the brain by the CYP450 isoform CYP2E1 [20]. As NAPQI is covalently bound to GSH, it depletes GSH in the brain and may aggravate oxidative stress. In rats, cortical neuronal death involving cytochrome c release and caspase 3 activation is induced by paracetamol at doses below those required to produce hepatotoxicity [25].

It is also relevant that paracetamol crosses both the placental barrier and the fetal blood–brain barrier and remains in the bloodstream of the infant for prolonged periods time [26,27], increasing the risks of altered development of the fetal brain.

## 3. Epidemiological Studies Investigating the Impact of Paracetamol on the Developing Brain

### 3.1. Paracetamol Use during Pregnancy

Paracetamol has been widely recommended for the treatment of pain and fever in pregnant women, and it is being estimated that about every other pregnant woman resorts to the use of paracetamol during pregnancy. However, there are a number of prospective cohort studies to suggest that intake of paracetamol increases the likelihood of autism spectrum disorder (ASD) and attention deficit hyperactivity disorder (ADHD) in the offspring (Table 1). The seminal analysis of the Danish National Birth Cohort comprised 64,322 pregnant women who had been recruited between 1996 and 2002 [28] and answered two telephone interviews before (at 12 and 30 weeks of gestation) and a further interview 6 months after delivery. There was a moderately increased risk of physician-diagnosed ADHD, prescription of ADHD medication, or parental reports of ADHD-like behavior at 7 years of age when the child was ever exposed to paracetamol before birth (average hazard ratios [HR] 1.37, 1.29 and 1.13, respectively). HR increased for all three outcome variables when children were exposed to paracetamol for more than 20 weeks (1.84, 1.53 and 1.46). Prenatal paracetamol was also associated with an increased risk of ASD with, but not without, hyperkinetic traits (HR 1.51 and 1.06, respectively) [29]. ASD risks were increased in children whose mothers had taken paracetamol during 3 trimesters (HR 1.77 and 1.25). In a small sub-cohort of 1491 children assessed at 5 years of age by trained psychologists, intrauterine paracetamol exposure was also associated with poorer cognitive [30] and attention scores [31].

A statistically significant association between intrauterine paracetamol exposure, as recalled by mothers, and a diagnosis of ADHD was also observed in the Nurses’ Health Study [33] and the Norwegian Mother and Child Cohort Study [37]. In the Norwegian study, however, no increased likelihood of ADHD was observed when paracetamol had been taken for less than eight days or in only one trimester. Notably, an association also emerged between paternal use of paracetamol and offspring ADHD while maternal paracetamol six months before pregnancy had no effect.

Various other studies have linked intrauterine paracetamol exposure, as recalled and reported by mothers, to results of questionnaires as proxy measures of ADHD [32,34,35,36,38,39,40,41,42,44,45]. Four sequential systematic reviews concluded that the evidence available suggests that the risk of ADHD and ASD is increased following prenatal paracetamol exposure [46,47,48,49].

There are, however, several points of concern [50,51]. First, the questionnaires used have poor internal and external validity, as they were developed as screening instruments rather than diagnostic tools. This adds to the heterogeneity of the results, and some studies using questionnaires indeed failed to detect a significant impact of intrauterine paracetamol [43,44]. Second, ADHD and ASD are partially heritable traits which may go undiagnosed in adults. This source of confounding is difficult to control for in epidemiological studies. In a sample of 7921 genotyped mothers participating in the Avon Longitudinal Study of Parents and Children (ALSPAC) study, maternal polygenic risk scores for ADHD (but not ASD) were slightly but significantly linked both to infections (odds ratio [OR] 1.11) and use of acetaminophen during late pregnancy (OR 1.11) [52]. However, in the Nurses’ Health Study II cohort that included 8856 children (721 with ADHD), only paracetamol use at the time of pregnancy was associated with childhood ADHD (OR 1.34), while there was not effect for paracetamol ingestions during periods 4 years before or 4 years after the pregnancy [33]. Third, fever is one of the leading indications for use of paracetamol, and fever during pregnancy itself has been associated with lower performance intelligence quotients [30], increased risks of ASD [53,54] and ADHD [55]. These associations were similar whether the woman had used paracetamol or not. In separate investigations, maternal infections during pregnancy have been associated with ASD in the offspring [56]. None of the studies accounted for maternal migraine which may be another important confounding indication [57]. Fourth, the epidemiological studies mentioned rely on maternal reports to quantitate paracetamol intake during pregnancy which may lead to exposure misclassification.

The last concern has been addressed by an analysis of public health insurance data from Taiwan [58] and cohort studies measuring perinatal, fetal, or neonatal paracetamol and paracetamol metabolites [59,60,61,62] (Table 2). The first approach reported a weak association between prescription of paracetamol during pregnancy and physician-diagnosed ADHD in the offspring [58]. However, paracetamol may be obtained without prescription, and paracetamol prescribed before pregnancy or to other household members may be taken by a pregnant woman encountering fever or pain. Attempts to measure paracetamol and its metabolites met with the challenge that paracetamol has become an almost universal component of human blood or urine [63]. Unchanged paracetamol was indeed detected in all 140 urine samples provided by mothers participating in the Swedish Environmental Longitudinal, Mother and child, Asthma and allergy study [59]; all 1180 maternal plasma obtained 1–3 after birth of women of the Boston Birth cohort [60]; and in all 996 cord blood samples of infants enrolled in the same cohort [61]. While neither raw nor log-transformed urinary paracetamol concentrations displayed a normal distribution [63], data of the Swedish cohort study demonstrated a linear association between log-transformed urinary paracetamol concentrations and mother-reported paracetamol use during mid-pregnancy [59], as well as a small impact of paracetamol intake (by maternal report or urinary concentration) on language development at 3 years of age. In the Boston Birth cohort, paracetamol burden according to blood samples obtained from mothers and infants was related to physician-diagnosed ADHD [60,61]. Neonatal meconium collected after birth may actually be the best way to capture prolonged intrauterine exposure to paracetamol and other drugs, as it accumulates chemicals from the fetal bile and the fetal urine passed into the amniotic fluid which is ingested by the fetus. In the Canadian Gestation and the Environment Cohort, paracetamol in meconium was unrelated to the children’s intelligence examined at 6–8 years of age [64] but showed a dose–response association with physician-diagnosed ADHD [62]. Each doubling of exposure increased the odds of ADHD by 10% among 345 children analyzed, 199 (57.7%) of whom had detectable paracetamol in meconium, and 33 (9.6%) were diagnosed with ADHD. In a subset of 48 children who underwent resting-state functional magnetic resonance imaging (MRI) at 9–11 years of age, paracetamol detected in meconium was linked to altered brain connectivity between fronto-parietal and default mode network nodes to sensorimotor cortex clusters, mediating an association of intrauterine paracetamol exposure and ADHD [62].

While in 2015 the Food and Drug Administration (FDA) continued to support the use of paracetamol for pain and fever during pregnancy [65], the Pharmacovigilance Risk Assessment Committee of the European Medicines Agency (EMA/PRAC/157165/2019) stated in 2019 that a large amount of data on pregnant women indicated neither malformative nor fetal/neonatal toxicity, while epidemiological studies on neurodevelopment in children exposed to paracetamol in utero showed inconclusive results. It recommended that paracetamol can be used during pregnancy if clinically needed, but it should be used at the lowest effective dose for the shortest possible time and at the lowest possible frequency [66]. The cautious stance of the regulatory authorities in the USA and Europe partially reflects the lack of safe alternatives to treat fever and pain in pregnant women. A very recently published consensus statement supported by 91 scientists, clinicians, and public health professionals recommends to implement specific actions to caution pregnant women at the beginning of pregnancy to forego paracetamol unless its use is medically indicated, to consult with a physician or pharmacist if they are uncertain whether its use is indicated and before using paracetamol on a long-term basis, and to minimize exposure by using the lowest effective dose for the shortest possible time [67].

### 3.2. Postnatal Use of Paracetamol in Term and Preterm Newborn Infants

Paracetamol and ibuprofen have equal efficacy for the treatment of fever in infants [24], and paracetamol has evolved into a cornerstone of effective pain relief in neonates. Paracetamol allows for reduced dosing of opioids and may be used to effectively treat moderate pain after surgery, while there is insufficient evidence to support its use for painful procedures [68,69]. Paracetamol given after assisted vaginal birth may even increase the response to subsequent painful procedures [70]. Paracetamol has been furthermore advocated for the pharmacological closure of a patent ductus arteriosus in very preterm infants, although its efficacy is not superior to oral ibuprofen [2,3]. Follow-up examinations of preterm infants at 2 and 5 years of age currently do not point to altered neurodevelopmental outcome following postnatal paracetamol administration [71,72,73], while analysis of data from a survey among parents on 1515 US children found an increased risk of ASD following postnatal paracetamol administration before age two in boys but not in girls [74]. As a matter of concern, none of the trials employing paracetamol for closure of a patent ductus arteriosus in preterm infants registered with www.clinicialtrials.gov list ADHD and ASD as secondary outcome.

## 4. Animal Models

In contrast to the loads of epidemiological studies analyzing the associations between gestational exposure to paracetamol and behavior in offspring, very few investigations have addressed this issue in experimental animals. Injecting pregnant rats day 15–19 of pregnancy with low-dose paracetamol (15 mg/kg body weight) has been shown to result in a large number of genes up- or down-regulated in fetal brains [75], while injecting pregnant mice on day 12.5 of pregnancy with paracetamol at dosages causing acute liver toxicity, as shown by elevated plasma alanine transferase concentrations, has been shown to reduce birth weight, decrease the frequency of hematopoietic stem cells in offspring liver [76], and result in greater severity of airway inflammation in grown-up animals [77], demonstrating long-lasting effects in offspring. However, no behavioral assessment has been reported in these experiments. Offspring of pregnant mice receiving paracetamol at 150 mg/kg/d by gavage from gestational day 7 to delivery did not display altered open field locomotor activity at 30 days of life [78]. However, male mice showed reduced sexual behavior associated with decreased neuronal number in the sexually dimorphic nucleus of the preoptic area [79]. After 350 mg/kg/d, there was impaired nest-seeking behavior, augmented stereotypy, and decreased rostral grooming in male animals, as well as reduced exploratory behavior in three-chamber sociability in both sexes [80,81].

In rodents, developmental phases of brain development that take place during the last trimester of pregnancy in humans are being observed during the first 7–10 days of life, allowing to employ newborn rat or mouse pups to study human fetal intrauterine events [82,83,84]. Administration of paracetamol (30 or 60 mg/kg body weight) to 3- or 10-day-old mice was shown to result in altered locomotor activity in response to a novel home environment and impaired spatial learning in adulthood [85]. Neonatal paracetamol also blunted the analgesic effect of paracetamol given to adult animals. Notably, exposure on day 19 of age had no long-lasting effects [86]. These data point to a critical time window during brain development that corresponds to the last trimester of pregnancy in humans. Male mice exposed on day 10 of life to a single dose of paracetamol (30 mg/kg body weight) did not differ from controls while mice receiving a repeat dose 4 h apart showed altered locomotor and rearing activity when tested as adults [87]. In a separate series of experiments, effects of paracetamol could not be prevented by co-administration of cysteine and mannitol as antioxidants [88]. Low-dose paracetamol (5 or 15 mg/kg/d) during pregnancy, followed by postnatal administration until 60 days of life, also evoked changes in behavior and reduced social interaction of grown-up animals [89].

## 5. Chronic Exposure to Ultra-Low Concentrations of Paracetamol

The wide use of paracetamol as an analgesic and antipyretic available without prescription translates into paracetamol becoming a constant ingredient of sewage water [90]. Human urine samples in developed countries contain paracetamol at low concentrations irrespective of active paracetamol intake [63]. The common presence of paracetamol in the aquatic environment has prompted investigations on the effects of low concentrations of paracetamol in evolutionary distant marine species. Changes in development, behavior, enzyme activities, and DNA methylation patterns have been observed in zebrafish larvae and embryos exposed for several days to paracetamol at concentrations as low as 5 µg/L [91]. In small planktonic crustaceans of the genus *Daphnia* (water flea), paracetamol at 40 µg/L was found to alter glutathione S-transferase activity and behavior (swimming distance) [92], while sea mussels (*Mytilus edulis*) show altered gene expression patterns even at 40 ng/L [93]. While these observations bear little direct relevance for the developing human brain, they demonstrate that minute amounts of paracetamol may exert biological responses in evolutionary distant species.

## 6. Concluding Remarks

An array of diverse epidemiological studies link (prolonged) intrauterine exposure of paracetamol to ADHD and ASD in offspring, while there is little evidence that paracetamol taken during pregnancy is associated with brain function and development in a more general sense. Epidemiological studies cannot answer the question of whether or not this association represents a causal interference or is mediated by unaccounted confounders. The (few) experimental investigations published to date do show an impact of paracetamol on immature rodent animals, but the precise mechanisms are unknown. Despite decades of use for fever and pain, the actions of paracetamol on neurons have only recently been studied on a molecular level, and future work will have to elucidate how paracetamol may interfere with the developing brain.

## Figures and Tables

**Figure 1 ijms-22-11156-f001:**
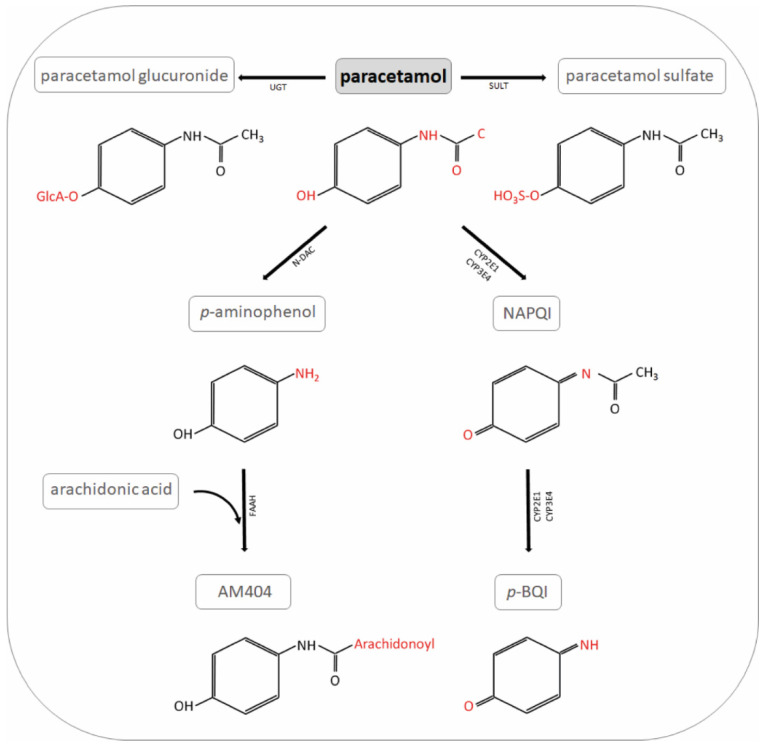
Paracetamol metabolites. NAPQI: *N*-acetyl-*p*-benzo-quinone-imine; UGT: UDP-glucuronyl transferase; SULT: sulfotransferase; N-DAC: *N*-deacetylase; GST: glutathione S-transferase; *p*-BQI: *p*-benzoquinone; FAAH: fatty acid amide hydrolase; AM404: *N*-arachidonoyl-phenolamine.

**Figure 2 ijms-22-11156-f002:**
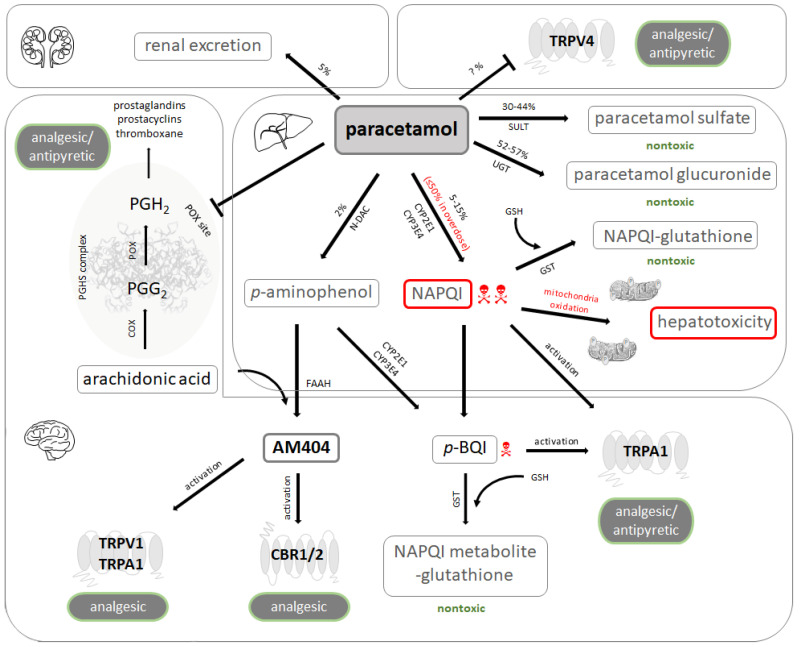
Paracetamol metabolism and pharmacology. Unchanged paracetamol (acetaminophen) is excreted in the urine to a small extent. It may also act as an antagonist of the vanilloid-subtype 4 receptors (TRPV4) in various tissues. Paracetamol is mainly metabolized in the liver. The metabolites generated by glucuronidation and sulfation are not toxic and subject to urinary excretion, while a minor fraction undergoes oxidative metabolism. This results in formation of *N*-acetyl-*p*-benzo-quinone-imine (NAPQI), a highly toxic intermediary produced by cytochrome P450 enzymes. NAPQI builds adducts with mitochondrial proteins and induces oxidative stress, nuclear DNA fragmentation, and subsequent cell necrosis. Paracetamol is becoming de-acetylated to *p*-aminophenol, which in turn is metabolized by the hepatic microsomal cytochrome P450 enzyme system to the toxic compound *p*-benzoquinone (*p*-BQI). Under physiological conditions, detoxification of NAPQI and *p*-BQI occurs by binding to glutathione (GSH) and subsequent renal excretion. The prostaglandin endoperoxide H synthase (PGHS) complex consists of a cyclooxygenase (COX) and a peroxidase (POX) moiety. Arachidonic acid is first transformed to the unstable prostaglandin G_2_ (PGG_2_) by COX, which is further reduced to prostaglandin H_2_ (PGH_2_) by POX. PGH_2_ gives rise to various endogenous regulators such as prostaglandins, prostacyclins, and thromboxane. Paracetamol induces analgesia and antipyresis by blocking prostaglandin synthesis at the POX site of PGHS complex. *p*-aminophenol undergoes conjugation with arachidonic acid by fatty acid amide hydrolase (FAAH) to *N*-arachidonoyl-phenolamine (AM404), which is a potent activator of TRPV1 and transient receptor potential ankyrin 1 (TRPA1) as well as a weak agonist of cannabinoid receptors type 1 and 2 (CBR1/2). Activation of these receptors by AM404 may mediate analgesic effects. NAPQI and *p*-BQI may also produce analgesic and antipyretic effects by activating TRPA1. UGT: UDP-glucuronyl transferase; SULT: sulfotransferase; N-DAC: *N*-deacetylase; GST: glutathione S-transferase.

**Table 1 ijms-22-11156-t001:** Paracetamol intake estimated based on maternal self-report.

Cohort	Country Code	Years of Birth	*n*	Age (Years)	Outcome	Assessed by	Association	References
Danish National Birth Cohort (DNBC)	DK	1996–2002	64,322	7	Behavior	Q D	Yes	[28]
					ADHD		Yes	
					ADHD			
					medication		Yes	
					Autism	D	Yes	[29]
Lifestyle During Pregnancy Study (DNBC sub-cohort)			1491	5	IQ	T	Yes	[30]
					Attention	T Q	Yes	[31]
			40,934	11	Behavior	Q	Yes	[32]
Nurses’ Health Study II	US	1993–2005	8856	≥8	ADHD	D	Yes	[33]
Craniofacial malformation, hemifacial microsomia study	US, CA	1996–2002	560	6–12	Behavior	Q	Yes/No	[34]
Norwegian Mother and Child Cohort Study (MoBa)	NO	1999–2008	15,256	3	Behavior	Q	Yes	[35]
							Yes	
							Yes	
			51,200	1½	Behavior	Q	Yes	[36]
			112,973	3–13	ADHD	D	Yes	[37]
			32,934	5	Behavior	Q	Yes	[38]
							Yes	
							Yes	
Auckland Birthweight Collaborative Study	NZ	1995–1997	871	7, 11	Behavior	Q	Yes	[39]
Avon Longitudinal Study of Parents and Children (ALSPAC)	UK	1991–1992	7796	7	Behavior	Q	Yes	[40]
			12,418	½–15	IQ Behavior	T	No	[41]
						Q	Yes	
INfancia y Medio Ambiente (INMA)	ES	2004–2008	2644	5	Behavior	T	Yes	[42]
Viva	US	1999–2002	1217	3	Cognition	T	No	[43]
Pelotas	BR	2015	3818	2	Cognition	T	No	[43]
		2004	3624	4	Behavior	Q	No	[44]
ALSPAC	UK	1991–1992	6200	7	Behavior	Q	Yes	[45]
Generation R	NL	2001–2005	3904	8				
INMA	ES	2004–2008	1513	4–5				
GASPII	IT	2003–2004	489	4				
DNBC	DK	1996–2002	61,430	7				
RHEA	GR	2007–2008	345	6				

Modes of outcome assessment: Q, questionnaire or structured interview (parents); T, test (psychologist, possibly computer-assisted); D, diagnosis (physician).

**Table 2 ijms-22-11156-t002:** Paracetamol intake measured by perinatal metabolites.

Cohort	Country Code	Years of Birth	*n*	Age (Years)	Outcome	Association	References
Swedish Environmental Longitudinal, Mother and child, Asthma and allergy (SELMA)	SE	2007–2010	754	3	Language	Yes	[59]
Boston Birth cohort	US	1998–2013	1180		Physician-diagnosed ADHD	Yes	[60]
		1998–2018	996	10	Physician-diagnosed ADHD	Yes	[61]
Gestation and the Environment Cohort (GESTE)	CA	2007–2009	195	6–8	IQ	No	[64]
	CA		345 48	6–7 9–11	Physician-diagnosed ADHD Functional MRI	Yes Yes	[62]

## Data Availability

Data sharing not applicable.

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
