# Peer review of "Paracetamol (Acetaminophen) and the Developing Brain"

_ijms, 2021, doi:10.3390/ijms222011156_

Round 1

Reviewer 1 Report

This review mainly provides a summary of selected epidemiological papers that deal with possible associations between ingestion of paracetamol (acetaminophen) in pregnancy and neonates and subsequent developmental disorders in offspring such as attention deficit hyperactivity disorder and autism spectrum disorder. It is not clear on what basis the papers were selected. A simple search of PubMed for “paracetamol ADHD” yields 64 papers.

Even allowing for the vagaries of PubMed and that a few may have been published after the submission of the authors review or overtaken by later publications, it would have been helpful to have an indication of why the papers reviewed were selected.

The papers reviewed are subdivided into two tables one listing papers based on maternal self-report of paracetamol ingestion, which as the authors indicate may lead to exposure misclassification. These are stated to be prospective which a key aspect in deciding on the value of such studies. Retrospective studies are usually of limited value because of the likelihood of selection bias. However, at least two of the studies in Table 1 are not prospective (ref, 37, 43), but others that are prospective are not included e.g. Avella-Garcia et al International Journal of Epidemiology, 2016, 1987–1996 doi: 10.1093/ije/dyw115.

The authors list a number of limitations of the studies in Table 1, but is often not clear which studies the limitations apply to. The study of Stergiakouli et l (2016, ref 37) is listed in Table 1, but not otherwise mentioned in the Review. This paper is notable only for its deficiencies which were highlighted in post publication correspondence in the journal:

JAMA Pediatr. 2017;171(4):394-395. doi:10.1001/jamapediatrics.2016.5034

JAMA Pediatr. 2017;171(4):395. doi:10.1001/jamapediatrics.2016.5037  

JAMA Pediatr. 2017;171(4):395-396. doi:10.1001/jamapediatrics.2016.5046

JAMA Pediatr. 2017;171(4):396. doi:10.1001/jamapediatrics.2016.5049

Given the ease with which modern data bases such as PubMed can be searched, a review is not particularly useful if it just gives incomplete lists of papers without giving a proper appraisal of their value and limitations.

Table 2 lists three studies in which paracetamol and/or its metabolites were measure in the perinatal period in meconium, maternal blood, urine and cord blood. These are potentially more useful as they provide evidence that paracetamol was actually consumed although when during pregnancy is difficult to ascertain with the methods and study design used.

There is a short section on animal models which one would hope might give insight into possible deleterious effects of paracetamol on brain development. However, several that I would have thought are relevant are not included. These include Thiele et al. (Am J Pathol.

2015; 185(10): 2805–2818) and Koehn et al. (F1000Res. 2020 Jun 8;9:573. doi: 10.12688/f1000research.24119.2). These papers describe effects of paracetamol in inducing immune/inflammatory responses in the uterus and placenta. Given the numerous reports of deleterious effects of inflammation on brain development (e.g. Hagberg et al. Nat Rev Neurol. 2015 Apr;11(4):192-208. doi: 10.1038/nrneurol.2015.13). I would have thought that these papers are relevant to this Review.

Because of the limitations listed above this review would only be a modest addition to the literature.

Author Response

The reviewer has a critical look at the epidemiological studies and animal experiments presented. While our manuscript considers itself as a narrative rather than a systematic review, the methodology behind is based on systematic tools, employing serial PubMed searches with appropriate keywords and a capture-recapture approach using the PubMed ‘cited by’ function. Being aware that we cannot be 100% sure not to have missed a cohort study, we consider this unlikely. The study by Avella-Garcia et al is listed in Table 1 (Ref. 38). It refers to the Spanisch INfancia y Medio Ambiente (INMA) study, part of which is also described in Ref. 44. Ref. 37 and 43 refer to the Avon Longitudinal Study of Parents and Children (ALSPAC). There is no reason to exclude this prospective study from this review.

We have included the animal experiments cited by the reviewer, being aware that they do not address any behavioral changes in the offspring. The experiments carried out in Victoria, Australia (Koehn 2020, ref 75) demonstrate that low-dose paracetamol given to pregnant rats causes up- or down-regulation of several genes in the fetal brain, while the experiments that were done in Hamburg, Germany (Thiele 2015, Ref 76, and Karimi 2015, Ref 77) employed high-dose paracetamol in pregnant mice that caused reversible liver damage in the dams, reduced birth weight in the offspring, and rendered them more susceptible to inflammatory airway disease when grown up. We also added updates of animal experiments carried out in Londrina, Brazil (Ref. 81) and Uppsala, Sweden (Ref. 85). These animal experiments point to increased susceptibility towards early paracetamol exposure of male, as opposed to female animals which has also been reported in humans (Ref 74). 

After submission of the manuscript (actually even after submission of the reviews), a consensus statement of 91 scientists, clinicians and public health care workers has been published in Nature Reviews Endocrinology, urging to implement specific action to caution pregnant women at the beginning of pregnancy to forego paracetamol unless its use is medically indicated, to consult with a physician and pharmacist before using paracetamol on a long-term basis, and to minimize exposure by using the lowest effective dose for the shortest possible time. While the association between paracetamol intake during pregnancy and subsequent ADHD and autism spectrum disorder is a matter of fierce debate (reflected in in the post publication history of several manuscripts, as noted by the reviewer, and contributions to a special issue of Paediatric Perinatal Epidemiology in 2020 [ref 49-51]), this consensus statement (added to the revised manuscript, ref 67) indicates that the topic is being viewed as a pressing issue that warrants putting together the puzzle of epidemiological, biochemical and experimental data pieces. We hope our manuscript can contribute to such endeavors.

Reviewer 2 Report

The paper by Bührer, C. is really interesting, comprehensive, and scientifically sound. I really think that it would attract a lot of readership for this journal. The only small comment I have got that it would be helpful if the authors explain the structure of the paracetamol and its metabolites through diagram. I have seen that the authors explained it even in the abstract but it would be interesting to look at the real structure and its metabolites though a diagram.

Author Response

We are thankful for this suggestion and have prepared a diagram showing the chemical structure of paracetamol and its metabolites. We chose to supply this is on a separate sheet as the figure is already quite busy.

Reviewer 3 Report

Ref: ijms-1380467

Title: Paracetamol (acetaminophen) and the developing brain

Minor review

Comments:

  1. Table 1 should be constructed more detailed. There is too many abbreviations. Also the symptoms are not specified e.g. “attention”.
  2. The animal model section should be described more detailed. There are papers on molecular basis on neurotoxicity of acetaminophen during neurodevelopment based on animal models.
  3. Although the subject of the work itself is not new, the review paper is mostly based on the latest data (56 of 81 reference list come from 2016-2021). This is a big advantage.

Author Response

  • We agree that the flurry of abbreviations in Table 1 has rendered it rather confusing. This had been mostly due to various tools to examine the behavioral outcome of the children (ASQ, CBBL, CPR-R, DAWBA, EAS, K‑CPT, SDQ). Virtually all studies relying on maternal self-report (which therefore have in common the problem of exposure misclassification) have been subject to four extensive systematic reviews published previously [ref. 45-48] and subject to individual criticism [ref. 49-50]. Therefore, we chose not to add more points but instead to make clear whether the outcome was based on a diagnosis of a physician, testing of the child by a professional (e.g. psychologist), or parental perceptions collected by questionnaires. This is pivotal to judge potential bias in endpoint misclassification, as the use of parent-completed questionnaires, which mostly have been developed as a screening tool, may provide questionable results. Their validity may be further impeded by collapsing various pseudo-continuous variables on an ordinal scale into dichotomous outcome estimates.
  • We updated and augmented the animal model section (ref 75-77, 80, 84) conceding that the issue does not appear to be well investigated experimentally.
  • This manuscript is focused indeed on recent developments, given that the epidemiological studies relying on maternal self-report have been subject to previous reviews and criticisms [ref. 46-50].